# Modeling of the Efficiency of the Centrifugal Conical Disk Dispenser of Bulk Materials

**DOI:** 10.3390/ma17081815

**Published:** 2024-04-15

**Authors:** Vasyl Dmytriv, Michał Bembenek, Vasyl Banha, Ihor Dmytriv, Damian Dzienniak, Saltanat Nurkusheva

**Affiliations:** 1Department of Design Machine and Automotive Engineering, Lviv Polytechnic National University, 79013 Lviv, Ukraine; vasyl.i.banha@lpnu.ua; 2Department of Manufacturing Systems, Faculty of Mechanical Engineering and Robotics, AGH University of Krakow, A. Mickiewicza 30, 30-059 Krakow, Poland; ddamian@agh.edu.pl; 3Department of Motor Vehicle Transport, Lviv Polytechnic National University, 79013 Lviv, Ukraine; ihor.v.dmytriv@lpnu.ua; 4Department of Organization of Transport, Traffic and Transport Operations, L. N. Gumilyov Eurasian National University, Satbaev 2, Astana 010000, Kazakhstan; saltanat.nurkusheva@kazatu.kz; 5Department of Transport Equipment and Technologies, S. Seifullin Kazakh Agrotechnical Research University, Zhenis 62B, Astana 010011, Kazakhstan

**Keywords:** centrifugal dispenser, productivity, conical disk, differential equations, the Runge–Kutta method, factorial experiment, correlation equation, bulk material, Fisher’s test

## Abstract

Centrifugal disk dispensers are widely used in various tasks of dosing bulk, dispersed materials. The design of the disk depends on the physical and mechanical characteristics of the dosing medium. The work discusses the development of an analytical model of the movement of a material particle along a conical centrifugal disk depending on the kinematic characteristics of the dosing process and the characteristics of the dosing material, as well as experimental confirmation of the theoretical model, which is relevant for the calculation and design of working elements of this type. The obtained system of differential equations is solved using the Runge–Kutta numerical method. Experimental studies were carried out using the method of a planned factorial experiment. The experiment was conducted for three factors at three levels. The feedback criterion was the performance of a centrifugal conical disk dispenser for bulk materials. The disk cone angle was set at 10, 20, and 30°. The disk diameter was 130, 150, and 170 mm, the gap between the disk and the edge of the hopper neck was 6, 8, and 10 mm, and the rotational speed of the conical disk was 0.65, 1.02, and 1.39 rad/s. The dispensing rate of the dispenser ranged from 15 to 770 g/s, depending on the values of the experimental factors. For use in the regression equation of the natural values of the factors, a method of transforming the terms of the equation from coded values to natural ones is provided. The obtained experimental correlation dependencies were checked for reproducibility with Cochrane’s test, and the adequacy of the model was checked using Fisher’s test. The significance of the coefficients in the correlation equation was evaluated using the Student’s *t*-test. The difference between the experimental data and the results of the theoretical modeling does not exceed 5%. The obtained system of differential equations makes it possible to model the radial velocity of the ascent of bulk material from the conical rotating disk depending on the rotation frequency, disk diameter, and the height of the annular gap between the discharge throat of the hopper and the conical disk. The analytical model enables the modeling of the productivity of the conical dispenser for bulk materials for arbitrary parameters of rotation frequency, disk diameter, and the size of the annular gap between the discharge throat of the hopper and the conical disk.

## 1. Introduction

Rotating disks are widely studied, and their operating parameters are modeled owing to their numerous applications in various industries and machines with rotating components, such as gas and steam turbines, pumps, turbogenerators, compressors, flywheels, ship propellers, and automotive braking systems (Gupta et al. [1], Hojjati and Hassani [2], Singh and Ray [3]). Centrifugal disks are also used in mechanical engineering for the finish machining of parts. Sutowski et al. [4,5] investigated the process of machining with a centrifugal disk using simulation to evaluate the distribution of kinetic energy of the working medium in the Ansys LS-DYNA software. Yıldırım [6] conducted a parametric study of the stress induced by centrifugal force in power-law-graded hyperbolic disks to assess their performance.

The centrifugal dosing method at the microdisk level has gained popularity In medical research. Madadelahi et al. [7] provided a review and analysis of centrifugal microfluidic platforms on disks that dose, separate, and mix various components, both liquid and solid, in microproportions.

The centrifugal method of dosing loose materials is also widely used [8,9,10,11,12]. The rotating disk can have additional blades of a given shape, which affects the dosing efficiency. Research on centrifugal dispensers encompasses a wide range of disk-based working elements. The complexity of the mathematical description of the dosing process leads to different options for abstracting the descent of the loose dispersed medium from the disk during its rotation. In particular, Villette et al. [13] investigated the centrifugal disk dispenser as a model of hybrid centrifugal spreading for the study of the spatial distribution of fertilizers and its assessment using the transverse coefficient of variation. At the same time, the accuracy, uniformity, and constancy of centrifugal spreaders have become the focus of large-scale research. Yinyan et al. [14] numerically modeled the performance of a centrifugal disk with the help of discrete element method (DEM) software. They modeled the influence of particle speed on the efficiency of centrifugal dosing and explained the change in dosing by the action of the inertial force on the particles during their departure from a high-speed centrifugal disk dispenser of bulk materials (fertilizers). The DEM method is widely used for modeling various technical and technological tasks, including oscillations that occur during dosing processes, which are analogous to vibrations of multilayer surfaces [15,16].

With the development of computer technology, DEM software for numerical modeling has found wide application in the field of agricultural engineering, allowing for the detailed tracking of particle movement. Especially worthy of notice here is the work of Zeng et al. [17], Song et al. [18], and Liu et al. [19]. The discrete element method is also used in the manufacturing industry. Bembenek et al. [20] modeled the briquetting process of fine-grained materials in a roller press using the discrete element method. The creation of a simulation model based on DEM software made it possible to analyze the performance of and numerically simulate the dosing process with centrifugal disk dispensers. Based on this research, studies of various operating parameters of dispensers were conducted to investigate the factors affecting the stability of the height of the dispersed bulk material; in particular, such studies were performed by Sun et al. [21] and Du et al. [22]. However, the study of the dynamics of fertilizer particles and their movement on the surface of the centrifugal disk under different operating parameters was not fully considered, and these studies were conducted as simulations using DEM methods.

Several factors affect the dosing accuracy of a centrifugal disk dispenser. Xu et al. [23] numerically predicted disk wear based on DEM modeling and the effect on dosage uniformity and stability. Moreover, Bembenek et al. [24] studied the wear mechanism of beaters of coal grinding mills. Of course, changing the size of the dosing mechanism affects the efficiency and accuracy of dosing. Villette et al. [25] investigated the influence of the speed of a fluted disk on the efficiency of the dosing of bulk material, in particular, fertilizer granules. Gao et al. [26] studied the movement of particles in a high-speed centrifugal for seed dosing using the DEM method. Yildirim [27] studied the influence of the cone angle and the disk rotation speed on the uniformity of fertilizer distribution in single-disk rotary fertilizer spreaders. Van Liedekerke et al. [28] used DEM to simulate a spinning disk fertilizer spreader, which allows for the simulation of the ascent of an individual particle. There is also a lack of a complete physical model, which prevents the analysis of the design of the technical system with the parameters of the dosing technological process. Kruszelnicka et al. [29] used DEM to model the grinding process to prove the abstraction of some parameters of the physics of the process. Karwat et al. [30] pointed out that when modeling the flow of bulk materials, it is important to correctly calibrate the input parameters on a macro scale; that is, it is more important to reflect the behavior of the entire mixture flow than the dependence between individual particles. Boikov et al. [31] also pointed out the importance of determining the behavior of the material in the macro dimension. They noted that in the case of spherical particles, seven modeling input parameters must be defined, which, if they vary linearly, can be described by an approximation function containing these parameters. The development of a reliable simulation model, as close as possible to the actual behavior of bulk materials, requires the correct calibration of the input parameters of the model, for example, the shape and size of the particles, coefficients of friction, and recovery or bulk density, which are determined experimentally in laboratory conditions [31,32,33].

It is worth noting that the properties of the material significantly affect the accuracy of the simulation. In their study of a disk dispenser, Barrios et al. [34] examined the interaction between fluted centrifugal disks and granules. They specifically focused on cases where the disks were made of polylactic acid (PLA) and the brushes were made of nylon. Artur [35] investigated the influence of disk rotation speed, material feeding position, and blade angle on the spatial distribution of bulk material (fertilizer). It has been demonstrated that changing the operating parameters can affect the allocation results to adapt to different operating requirements. Statsenko et al. [36] experimentally investigated the performance of a disk dispenser for bulk materials. An et al. [37] modeled in Ansys Fluent the flight of grains from a disk-wicking device depending on the technical and technological parameters of the disk. Symons [38] studied the flow of wet granular material along the inner surface of a conical disk during its rotation and modeled the trajectories of the grain depending on its moisture content. Cool et al. [39] proposed a three-dimensional ballistic model and investigated the influence of the rotation of a concave disk on the flight path of fertilizer grains in the air and their subsequent position when they fall to the surface. The influence of material properties on the dosing process was also confirmed by research by Karwat et al. [40]. Based on the actual mass and power efficiency values obtained in a laboratory experiment, they verified the theoretical design methods and the results of the model of the numerical discrete element method.

The above-listed studies are limited by the number of parameters that can be taken into account during the modeling of the dosing process and in experimental research. To take into account the maximum number of indicators, Dmytriv et al. [41] propose using the method of complex criterion equations that combine several parameters in experimental studies.

Several issues regarding the substantiation of material movement parameters on conical rotating surfaces have not been investigated, while the radius of the disk from the angle of the cone, rotational frequency, and energy indicators of material dosing have not been substantiated, and the optimal structural and kinematic parameters of the conical dosing disk depending on technological factors have not been established.

Therefore, the development of an analytical model of the movement of a particle of material along a conical disk, depending on the kinematic characteristics of the dosing process and the characteristics of the dosing material and its experimental confirmation, are relevant for the calculation and design of working elements of this type. Such dosing systems are widely used for dosing and mixing processes in pharmacy, the chemical industry, food production, the industries of processing materials and mixtures, as well as in agricultural production [42].

The purpose of this research was to develop a mathematical model of the productivity of a bulk materials dispenser at various structural, technological, and operational parameters of a conical centrifugal dosing body and experimental verification of the theoretical model.

## 2. Materials and Methods

The aim of the study is to substantiate the design parameters and operating modes of the conical working element of the dispenser (the diameter of the dosing working element—the diameter of the conical disk, the annular gap between the discharge throat of the hopper and the conical disk, and the rotational speed of the conical disk), which affect dosing productivity. The angle of the conical dosing disk was a constant parameter. The bulk material is real. It is a mixture of crushed grains, flour (as a mixture of crushed grains, including corn, buckwheat, and others). The material mixture was selected so that its density was 550 kg/m^3^. The moisture content of the bulk material (mixture) was 13%, the angle of natural repose was 33°, the coefficient of sliding friction of the bulk material *f* = 0.2–0.25, the coefficient of internal sliding friction (adhesion) *f*_1_ = 0.65, and the external coefficient of sliding friction between the bulk material and the conical dosing disk *f*_2_ = 0.443.

### 2.1. Analytical Model of the Productivity of a Centrifugal Conical Disk Bulk Material Dispenser

In order to develop an analytical model of productivity, let us consider the movement of a material point of dispersed materials on the surface of a conical dosing working body, the generator of which is inclined at an angle *α* at its base. The position of the moving material point can be specified using two parameters: *l*—the distance from the top of the conical dosing working body to point *M* (*l* = *OM*), and the dihedral angle *φ* between some fixed (immovable) plane and the moving plane that passes through the axis of the conical dosing working body and point *M* (Figure 1).

The absolute velocity vector of the material point *M* is written in the form:(1)$$\vec{\vartheta }=\vartheta_{l}\cdot {\vec{l}}_{0}+\vartheta_{\varphi }\cdot \vec{\tau }.$$
where$\vartheta_{l}=dl/dt=\dot{l}$—radial component of velocity (velocity along the generator) [m/s];${\vec{l}}_{0}$—a unit vector directed along the generating *OM*;$\vartheta_{\varphi }$—transverse component of velocity [m/s]:



(2)
$$\vartheta_{\varphi }=\frac{d\varphi }{dt}\cdot l\cdot \cos\alpha ;$$

*l*—distance from the top of the conical dosing working body to point *M* [m];$\vec{\tau }$—a unit vector directed along the tangent to the parallel (perpendicular to ${\vec{l}}_{0}$ and *O*_1_*M*);$d\varphi /dt=\dot{\varphi }$—angular speed of rotation of the moving plane [rad/s];*α*—angle of the truncated cone at its base [°].


We determine the acceleration of point *M*, and we differentiate relationship (1) concerning time, taking into account Equation (2):(3)$$\vec{a}=\frac{d\vec{\vartheta }}{dt}=\frac{{d\vartheta }_{l}}{dt}\cdot {\vec{l}}_{0}+\vartheta_{l}\cdot \frac{d{\vec{l}}_{0}}{dt}+\left(\frac{d^{2}\varphi }{{dt}^{2}}\cdot l+\frac{d\varphi }{dt}\cdot \vartheta_{l}\right)\cdot \cos\alpha \cdot \vec{\tau }+\frac{d\varphi }{dt}\cdot l\cdot \frac{d\vec{\tau }}{dt}\cdot \cos\alpha .$$

Let us introduce the angular velocity vector:(4)$$\vec{\theta }=\frac{d\varphi }{dt}\cdot \vec{K},$$
where $\vec{K}$ is a unit vector directed along the axis of the Oz cone.

Taking into account the known dependencies of the kinematics, we obtain:(5)$$\frac{d{\vec{l}}_{0}}{dt}=\vec{\theta }\times {\vec{l}}_{0}=\frac{d\varphi }{dt}\cdot \cos\alpha \cdot \vec{\tau },$$
(6)$$\frac{d\vec{\tau }}{dt}=\vec{\theta }\times \vec{\tau }=\frac{d\varphi }{dt}\cdot \vec{n}.$$

Substituting expressions (5) and (6) into Equation (3), we obtain the relationship for the acceleration vector of the material point *M*:(7)$$\vec{a}=\frac{{d\vartheta }_{l}}{dt}\cdot {\vec{l}}_{0}+2\cdot \vartheta_{l}\cdot \frac{d\varphi }{dt}\cdot \cos\alpha \cdot \vec{\tau }+\frac{d^{2}\varphi }{{dt}^{2}}\cdot l\cdot \cos\alpha \left(\vec{\tau }+\vec{n}\right).$$

Let us separate from the moving dispersed material, adjacent to the surface of the conical dosing working body, which rotates with an angular velocity ω, an elementary volume with a height h and a square area $S_{0}$ in a plan view (Figure 2). This volume will experience forces during its movement, as indicated in Figure 2.

Let us write down these forces along with their definitions.

The weight of the elementary volume is determined by the formula
(8)$$G=\rho \cdot g\cdot h\cdot S_{0},$$
whereρ—density of the material [kg/m^3^];g—gravitational acceleration [m/s^2^];*h*—height of the elementary volume of the dispersed material [m];$S_{0}$—cross-sectional area of the volume of the dispersed material in a plan view [m^2^].

The pressure of the bulk material acting on the elementary volume located on the conical disk is determined by the formula, assuming its constant value:(9)$$\vec{p}=\frac{\rho \cdot g\cdot D_{K}}{8\cdot f\cdot \tan\beta }\;,$$
where$D_{K}$—the diameter of the disk above which the bulk material is located;*f*—coefficient of friction of the material sliding against the walls of the bunker;*β*—angle of packing of particles of the bulk material [°].

The normal reaction from the weight of the elementary volume of the material on the surface of the conical disk is denoted by $\vec{N}$.

The friction force of the elementary volume of the bulk material on the surface of the conical dosing working body is determined by the formula
(10)$$F_{2T}=f_{2}\cdot N,$$
where $f_{2}$ is the external sliding friction coefficient between the elementary volume of the bulk material and the surface of the conical dosing working body.

This force is directed opposite to the relative velocity vector, which is determined by the formula
(11)$${\vec{\vartheta }}_{rel}=\vartheta_{l}\cdot {\vec{l}}_{0}+l\cdot \cos\alpha \cdot \left(\frac{d\varphi }{dt}-\omega \right)\cdot \vec{\tau },$$
where *ω* is the angular velocity of rotation of the working body of the conical disk, in rad/s.

The force of internal sliding friction, which is directed opposite to the absolute velocity vector, is determined by the formula
(12)$$F_{1T}=f_{1}\cdot p\cdot \cos\alpha \cdot S_{0}\;,$$
where $f_{1}$ is the coefficient of internal sliding friction (adhesion).

The absolute velocity vector of the material point *M* is determined by the formula
(13)$$\vec{\vartheta }=\vartheta_{l}\cdot {\vec{l}}_{0}+\frac{d\varphi }{dt}\cdot l\cdot \cos\alpha \cdot \vec{\tau }.$$

Considering the selected elementary volume of the bulk material as a material point, we formulate the differential equation of its motion. According to the principle of superposition of forces, it is possible to write:(14)$$m\cdot \vec{a}=\vec{G}+{\vec{F}}_{1T}+{\vec{F}}_{2T}+\vec{N}+\vec{p}\cdot S_{0},$$
where $m=\rho \cdot h\cdot S_{0}.$ We project the vector equality (14) on axes ${\vec{l}}_{0}$, $\vec{\tau }$, and ${\vec{n}}_{1}$, taking into account expression (7) for acceleration and the remark about the directions of forces:(15)$$\left\{\begin{array}{c} S_{0}\cdot \rho \cdot h\cdot \left(\frac{d\vartheta_{l}}{dt}-\frac{d^{2}\varphi }{{dt}^{2}}\cdot l\cdot {\left(\cos\alpha \right)}^{2}\right)=\left(p\cdot S_{0}+G\right)\cdot \sin\alpha -p{\cdot S}_{0}\cdot \cos\alpha \cdot f_{1}\cdot \frac{\vartheta_{l}}{\left|\vartheta \right|}-f_{2}\cdot N\cdot \frac{\vartheta_{l}}{\left|\vartheta_{rel}\right|};\;\;\;\\ S_{0}\cdot \rho \cdot h\cdot \left(\frac{d^{2}\varphi }{{dt}^{2}}\cdot l+2\cdot \frac{d\varphi }{dt}\cdot \vartheta_{l}\right)\cdot \cos\alpha =-p{\cdot S}_{0}\cdot {\left(\cos\alpha \right)}^{2}\cdot f_{1}\cdot \frac{\frac{d\varphi }{dt}\cdot l}{\left|\vartheta \right|}-f_{2}\cdot N\cdot \frac{\left(\frac{d\varphi }{dt}-\omega \right)\cdot l\cdot \cos\alpha }{\left|\vartheta_{rel}\right|};\\ S_{0}\cdot \rho \cdot h\cdot \frac{d^{2}\varphi }{{dt}^{2}}\cdot l\cdot \cos\alpha \cdot \sin\alpha =-N+\left(p\cdot S_{0}+G\right)\cdot \cos\alpha ,\;\;\;\;\;\;\;\;\;\;\;\;\;\;\;\;\;\;\;\;\;\;\;\;\;\;\;\;\;\;\;\;\;\;\;\; \end{array}\right.$$
where$\left|\vartheta \right|$, $\left|\vartheta_{rel}\right|$—the absolute and relative speed of movement of a material particle of dispersed materials:


(16)
$$\left|\vartheta \right|=\sqrt{{\vartheta_{l}}^{2}+{\left(\frac{d\varphi }{dt}\cdot l\cdot \cos\alpha \right)}^{2}},$$



(17)
$$\left|\vartheta_{rel}\right|=\sqrt{{\vartheta_{l}}^{2}+{\left(\frac{d\varphi }{dt}-\omega \right)}^{2}\cdot l^{2}\cdot {\left(\cos\alpha \right)}^{2}}.$$


From the last equation in the system of Equation (15), we obtain the equation for determining the normal reaction *N*:(18)$$N=\left(p\cdot S_{0}+G\right)\cdot \cos\alpha -S_{0}\cdot \rho \cdot h\cdot \frac{d^{2}\varphi }{{dt}^{2}}\cdot l\cdot \cos\alpha \cdot \sin\alpha =S_{0}\cdot N_{0},$$
where $N_{0}$ is the reaction acting per unit area:$$N_{0}=\left(p+\rho \cdot h\cdot g\right)\cdot \cos\alpha -\rho \cdot h\cdot \frac{d^{2}\varphi }{{dt}^{2}}\cdot l\cdot \cos\alpha \cdot \sin\alpha .$$

The first two differential equations of the second order of the system of Equation (15) are for finding quantities: $\vartheta_{l}$, dφ/dt, *l*, *φ.*

The system of differential equations is solved using the Runge–Kutta numerical method. We reduce the system of Equation (15) to the form:(19)$$\left\{\begin{array}{c} \frac{dl}{dt}=\vartheta_{l};\;\frac{d\varphi }{dt}=u;\;\;\\ \frac{d\vartheta_{l}}{dt}=u^{2}\cdot l\cdot {\left(\cos\alpha \right)}^{2}+\frac{p+\rho \cdot h\cdot g}{\rho \cdot h}\cdot \sin\alpha -\frac{p\cdot \cos\alpha }{\rho \cdot h}\cdot f_{1}\cdot \frac{\vartheta_{l}}{\left|\vartheta \right|}-\frac{f_{2}\cdot N_{0}}{\rho \cdot h}\cdot \frac{\vartheta_{l}}{\left|\vartheta_{rel}\right|};\\ \frac{du}{dy}=\frac{2\cdot u\cdot \vartheta_{l}}{l}-\frac{1}{\rho \cdot h}\left(p\cdot {\left(\cos\alpha \right)}^{2}\cdot f_{1}\cdot \frac{u}{\left|\vartheta \right|}+f_{2}\cdot N_{0}\cdot \frac{\left(u-\omega \right)\cdot l\cdot \cos\alpha }{\left|\vartheta_{rel}\right|}\right).\;\;\;\;\;\; \end{array}\right.$$
where *u* is the angular speed of rotation of the plane, in rad/s.

We solve the system of Equation (19) under the initial conditions:$$t=0;\;\varphi =0;\;u=0;\;l=l_{0};\;\vartheta_{l}=\vartheta_{0}.$$

We integrate the system of Equation (19) if the condition is fulfilled $l<\left[D_{K}/\left(2\cdot \cos\alpha \right)\right]$. When the condition $l=\left[D_{K}/\left(2\cdot \cos\alpha \right)\right]$ is fulfilled, the obtained value of the radial velocity $\vartheta_{l}$ is taken to determine the performance of the centrifugal conical disk dispenser according to the formula
(20)$$Q_{K}=\pi \cdot D_{K}\cdot h\cdot \rho \cdot \vartheta_{l}.$$
where$D_{K}$—diameter of the disk above which there is bulk material [mm];*h*—height of the elementary volume of the bulk material, corresponding to the height of the annular gap between the discharge neck of the hopper and the dosing cone disk [mm];ρ—density of loose material [g/mm^3^];$\vartheta_{l}$—radial velocity of the loose material at the exit from the disk [mm/s].

### 2.2. Factorial Planned Experiment

The purpose of the planned factorial experiment was to check the conformity of the analytical modeling of the performance of the centrifugal conical disk dispenser of bulk materials with the data of experimental studies.

The research was conducted as a planned factorial experiment. The basis is the method of experimenting [43], taking into account the importance of each factor and decoding the members of the regression dependence into the coefficients of the equation for the natural values of the factors. The response criterion was the performance parameter of the centrifugal conical disk dispenser *Q_K_*, in g/s, the factors were the rotation frequency *n*, rad/s—*x*_1_, the height of the annular gap between the outlet of the hopper and the conical dosing disk *h*, mm—*x*_2_, and the diameter of the conical disk *D_K_*, mm—*x*_3_. Factor values and coded coefficients, according to the theory of experimental design, are given in Table 1.

To study the influence of these factors, according to the matrix of the experiment (Table 2), the experiment was carried out three times according to the previously mentioned method [43].

The experiment was carried out on a stand for experimental research in laboratory conditions. The parameter values were measured by intelligent sensors. The digital code from the sensors was read by a computer through the data transmission-reception interface. The block diagram of the dispenser and the general view of the laboratory installation are shown in Figure 3.

Based on Table 2, we calculate the coefficients of the regression equation. The values of the regression coefficients characterize the contribution of each factor to the value of the response function and are calculated according to the following formulas:(21)$$b_{1}=\frac{\sum \left(x_{1}\cdot y\right)}{18},\;b_{2}=\frac{\sum \left(x_{2}\cdot y\right)}{18},\;b_{3}=\frac{\sum \left(x_{3}\cdot y\right)}{18},\;b_{11}=\frac{\sum \left({\left(x_{1}^{/}\right)}^{2}\cdot y\right)}{6},\;b_{22}=\frac{\sum \left({\left(x_{2}^{/}\right)}^{2}\cdot y\right)}{6},\;b_{33}=\frac{\sum \left({\left(x_{3}^{/}\right)}^{2}\cdot y\right)}{6},\;b_{12}=\frac{\sum \left(x_{1}\cdot x_{2}\cdot y\right)}{12},\;b_{13}=\frac{\sum \left(x_{1}\cdot x_{3}\cdot y\right)}{12},\;b_{23}=\frac{\sum \left(x_{2}\cdot x_{3}\cdot y\right)}{12},\;\;b_{123}=\frac{\sum \left(x_{1}\cdot x_{2}\cdot x_{3}\cdot y\right)}{8},\;b_{0}=\frac{\sum y}{27}-0.67\cdot b_{11}-0.67\cdot b_{22}-0.67\cdot b_{33}.$$

The results of calculating the coefficients of the regression equation are given in Table 3, Table 4 and Table 5, for a conical disk with a generating angle of the cone at its base of 10°, 20°, and 30°, respectively.

To use the natural values of the factors in the regression equation, we convert the linear terms of the equation from coded values to natural values, which were determined by the formula
(22)$$b_{i}\cdot x_{i}=\frac{b_{i}}{\varepsilon_{i}}\cdot X_{i}-\frac{b_{i}}{\varepsilon_{i}}\cdot X_{i0},$$
where*X_i_*—natural value of the factor;*X_i_*_0_—natural value of the factor at zero level;ε—variation interval.

The transformation of the interacting linear terms of the equation was performed according to the formula
(23)$$b_{ij}\cdot x_{i}\cdot x_{j}=\frac{b_{ij}}{\varepsilon_{i}\cdot \varepsilon_{j}}\cdot \left(X_{i}\cdot X_{j}-X_{i}\cdot X_{j0}-X_{j}\cdot X_{i0}+X_{i0}\cdot X_{j0}\right),$$

The transformation of the quadratic terms was performed using the formula
(24)$$b_{ij}\cdot x_{i}^{2}=\frac{b_{ii}}{\varepsilon_{i}^{2}}\cdot \left(X_{i}^{2}-2\cdot {X_{i}\cdot X}_{i0}+X_{i0}^{2}\right),$$

The results of the calculation of the natural coefficients of the regression equation are given in Table 3, Table 4 and Table 5, for a conical disk with a cone generating angle at its base of 10°, 20°, and 30°, respectively.

## 3. Results and Discussion

### 3.1. Results of Modeling Dosing Efficiency

In order to establish the influence of the design, technological, and operating parameters of the conical dosing working body on its performance *Q_k_*, a simulation was performed, after previously determining the speed of the flow of the bulk material from the system of Equation (19) using the Runge–Kutta method.

The density of the material was taken as $\rho =550\;kg/m^{3}$ or $\rho =0.55\cdot 10^{-3}\;g/{mm}^{3}$. The height of the annular gap between the discharge neck of the hopper and the metering conical disk was within *h* = 6…10 mm, the rotational frequency of the dosing conical disk was *n =* 0.65…1.39 rad·s^−1^, and the diameter was within *D_K_ =* 130…170 mm. The fixed values of the angle *α* of the generating cone at its base were 10°, 20°, and 30°_._

The simulation results are shown in Figure 4, Figure 5 and Figure 6.

Based on these values, the simulation results show that the dosing performance can vary within the range of 12 up to 800 g/s.

At the angle *α* of the generating cone at its base 10° (Figure 4), the frequency of rotation of the conical disk from 0.65 to 1.39 rad/s and the diameter of the disk from 130 to 170 mm, the dosing performance will vary from 12 to 210 g/s. The maximum dosing efficiency of the centrifugal conical disk dispenser will be achieved with a disk diameter of 170 mm, a disk rotation frequency of 1.39 rad/s, and a height of the annular gap between the discharge throat of the hopper and the dosing conical disk of 10 mm, and will amount to 210 g/s. With the same disk diameter and the height of the annular gap between the outlet neck of the hopper and the dosing cone disk, the minimum dosing performance of the centrifugal cone disk dispenser will be 120 g/s at a minimum rotation frequency of 0.65 rad/s.

As the diameter of the disk and the height of the annular gap between the discharge throat of the hopper and the dosing cone disk decrease, the dosing efficiency of the centrifugal cone disk dispenser decreases.

So, with a disk diameter of 130 mm, the height of the annular gap between the outlet neck of the hopper and the dosing conical disk of 6 mm, the maximum dosing performance will be 32.5 g/s at a disk rotation frequency of 1.39 rad/s, at a disk rotation frequency of 0.65 rad/s and the same design parameters of the dispenser, the minimum productivity will be 12.1 g/s, respectively.

At the angle *α* of the generating cone at its base 20° (Figure 5), the frequency of rotation of the conical disk from 0.65 to 1.39 rad/s, and the diameter of the disk from 130 to 170 mm, the dosing efficiency will vary from 20.3 to 340 g/s.

The maximum dosing performance of the centrifugal conical disk dispenser will be achieved with a disk diameter of 170 mm, a disk rotation frequency of 1.39 rad/s, and a height of the annular gap between the discharge throat of the hopper and the dosing conical disk of 10 mm, amounting to 340 g/s (Figure 5). With the same disk diameter and height of the annular gap between the discharge throat of the hopper and the dosing conical disk, the minimum dosing productivity of the centrifugal conical disk dispenser will be 167.2 g/s at a minimum rotation frequency of 0.65 rad/s.

Therefore, with a disk diameter of 130 mm and a height of the annular gap between the discharge throat of the hopper and the dosing conical disk of 6 mm, the maximum dosing productivity will be 42.6 g/s at a disk rotation frequency of 1.39 rad/s. At a disk rotation frequency of 0.65 rad/s and with the same structural parameters of the dispenser, the minimum productivity will be 20.3 g/s (Figure 5).

Figure 6 shows the modeling results for the α angle of the truncated cone at its base of 30°, the rotation frequency of the conical disk ranging from 0.65 to 1.39 rad/s, the disk diameter ranging from 130 to 170 mm, and the height of the annular gap between the discharge throat of the hopper and the conical disk of 6, 8, and 10 mm.

For the α angle of the truncated cone at its base of 30° (Figure 6), with the rotation frequency of the conical disk ranging from 0.65 to 1.39 rad/s and the disk diameter ranging from 130 to 170 mm, the dosing productivity will vary from 194 to 778.7 g/s. The maximum dosing productivity of the centrifugal conical disk dispenser will be achieved with a disk diameter of 170 mm, a disk rotation frequency of 1.39 rad/s, and a height of the annular gap between the discharge throat of the hopper and the dosing conical disk of 10 mm, amounting to 778.7 g/s. With the same disk diameter and height of the annular gap between the discharge throat of the hopper and the dosing conical disk, the minimum dosing productivity of the centrifugal conical disk dispenser will be 376.9 g/s at a minimum rotation frequency of 0.65 rad/s.

So, with a disk diameter of 130 mm and a height of the annular gap between the throat of the hopper and the dosing conical disk of 6 mm, the maximum dosing productivity will be 237.2 g/s at a disk rotation frequency of 1.39 rad/s. At a disk rotation frequency of 0.65 rad/s and the same structural parameters of the dispenser, the minimum productivity will be 113.6 g/s.

### 3.2. Results of Experimental Studies

According to the methodology, the real and natural values of the regression equations were calculated, which are shown in Table 3, Table 4 and Table 5.

With the decoded values of the coefficients of the regression equation, the dosing productivity model of the centrifugal conical disk dispenser for the angles of the truncated cone *α* to its base 10°, 20°, and 30°, respectively, will take on a natural form:(25)$$Q_{K}=165.5043+106.1478\cdot n+23.1952\cdot h-3.4021\cdot D_{K}+24.9371\cdot n\cdot h-1.4815\cdot n\cdot D_{K}+\;+0.0105\cdot h\cdot D_{K}-30.4071\cdot n^{2}-1.3918\cdot h^{2}+0.442\cdot D_{K}^{2}+0.2961\cdot n\cdot h\cdot D_{K},$$
(26)$$Q_{K}=164.5176+144.4081\cdot n-15.5005\cdot h-1.7246\cdot D_{K}-7.7388\cdot n\cdot h+0.5931\cdot n\cdot D_{K}+\;+0.0045\cdot h\cdot D_{K}-28.1833\cdot n^{2}-2.0002\cdot h^{2}+0.0621\cdot D_{K}^{2}+0.4232\cdot n\cdot h\cdot D_{K},$$
(27)$$Q_{K}=-2815.578+6830.101\cdot n+577.7693\cdot h+1.4054\cdot D_{K}-607.203\cdot n\cdot h-4.9544\cdot n\cdot D_{K}+\;+0.0867\cdot h\cdot D_{K}-654.011\cdot n^{2}-37.1962\cdot h^{2}-3.2687\cdot D_{K}^{2}+4.3652\cdot n\cdot h\cdot D_{K}.$$

The regression equations of the dependence of the dispenser productivity on the factors are presented graphically in Figure 7, Figure 8 and Figure 9, respectively, for the regression dependencies (25), (26), and (27).

The correctness of the experiment and the obtained experimental data were verified. The experiment was repeated three times at the same levels of factors. The results of the analysis of experimental data for the reproducibility of the experiments according to the Cochrane test, on the significance of the coefficients of the regression equations according to Student’s *t*-test, and the assessment of the adequacy of the models according to the Fisher *F*-test showed the following:

The calculated values of the Cochran test for confirming the reproducibility of the experiments, as per the dependencies (25), (26), and (27) governing the productivity of the bulk material dispenser, are 0.178, 0.113, and 0.0599, respectively. Each of these values is less than the tabular value of 0.2354.

The average value of the dispersion and reproducibility is *S^2^* = 0.0933.

To check the suitability of the regression Equations (25)–(27) for the characteristic description of the dependence of the optimization criterion on factors, we determine the Fisher criterion (*F*-criterion).

The adequacy variance $S_{ad}^{2}$ and the calculated value of the *F*-test are $S_{ad}^{2}$ = 0.003528 and *F_k_* = 1.688, respectively.

The tabular value of the *F*-test, taken for the calculated degrees of freedom of the main variance and the adequacy variance, respectively, is *F_t_* = 2.29.

We evaluate the adequacy of the model by the condition *F_r_* ≤ *F_t_*, respectively 1.688 ≤ 2.29; the models described by Equations (25)–(27) are adequate.

The analysis of the research results shows that the angle of the truncated cone of the disk to its base significantly affects its efficiency, and so does the rotation frequency of the disk.

The height of the annular gap between the discharge throat of the hopper and the conical disk is not a very crucial parameter; therefore, it can be considered dependent on the angle of the natural slope of the bulk material, and it affects the spontaneous discharge of the bulk material from the throat of the hopper. Therefore, this parameter is limited and is determined by the physical and mechanical characteristics of the bulk material. The diameter of the disk affects the speed of flow of the bulk material from the disk during its rotation. An increase in diameter leads to an increase in speed.

Taking into account the complexity of the analytical solution of the mathematical model, as well as conducting a physical experiment, researchers also use DEM numerical modeling [4,5,6,7]. However, numerical modeling does not allow for assessing the similarity of the obtained results with the physical process. Various researchers [8,9,10,11,12,13] believe that friction coefficients depend on speed, so its measurement directly in experimental studies can be somewhat overestimated. Therefore, it is important to study the tangential forces acting on the particles interacting with the vanes on the disk, and the parameters of the forces depending on the speed of the particle. To study the process of dosing bulk materials, a number of researchers take the movement of one particle on a flat disk with radially placed vanes [14,15,16,17,18,28]. Individual particle trajectories are explained by an initial position of the particle closer to the center of the disk, a lower initial outward radial velocity of the particle in the disk, or a higher counterforce on the particle moving along the disk, leading to a faster particle velocity and hence a later exit from the disk. In order to confirm the results of numerical simulation, Gao et al. [26] developed an experimental setup and studied the movement of a single particle, not a mixture. Therefore, the assessment of the physical process of bulk flow dosing is also impossible here. Yildiran [27] investigated the effect of cone angle and disk rotation speed on the uniformity of fertilizer distribution in single-disk rotary fertilizer spreaders. The cone of the disk was in the inner space, and blades were installed on the cone. The experiments were conducted with a flat disk and two conical disks, three speeds of disk rotation, and three rates of fertilizer. The disk cone angles were 0° (flat disk), 10°, and 20°. The coefficient of variation ranged from 9 to 43%, and the relative error also exceeded 10%. Yinyan et al. [14], studying a dispenser in the form of a flat disk with vanes that change the angle relative to the radius, conducted a comparison of numerical modeling with the results of a production experiment. The average coefficient of variation was the lowest (14.39%) for a single-row flow rate of 300 g/s, a blade angle of 15°, a spreader disk height of 95 cm, and a rotation speed of 600 r/min. Production verification tests showed that the average coefficient of variation relative to the effective spread width of the applicator was 16.74%. The relative error was more than 10% with respect to the simulation results; thus, the simulation model accuracy was relatively good.

However, the absence of an active element on the surface of the disk reduces the energy of the particles of the bulk material, which also reduces its kinetic energy. The presence of vanes on the surface of the disk will limit the movement of the material along the disk along two plane coordinates, and there will be only movement along one coordinate, which is directed along the generating disk.

Therefore, it is advisable to carry out similar studies on the design of a disk with blades.

## 4. Conclusions

Based on the results of theoretical modeling obtained from the experimental studies, the following conclusions can be drawn:

The results of the experimental studies agree with the results of the theoretical simulations. The difference between the data of experimental studies and the results of theoretical modeling does not exceed 5%.

At an angle of the truncated cone at its base of 30°, a rotation frequency of the conical disk 0.65 rad/s, a diameter of the disk from 130 mm, and a height of the annular gap between the discharge throat of the hopper and the dosing conical disk 10 mm, the theoretical dosing efficiency is 180.1 g/s, and the empirical value is 171.5 g/s. With a disk diameter of 170 mm, a disk rotation frequency of 1.39 rad/s, and a height of the annular gap between the discharge throat of the hopper and the dosing cone disk of 10 mm, the theoretical dosing efficiency is 778.7 g/s, and during the experiment, it was 745.4 g/s. With the same disk diameter and the height of the annular gap between the discharge throat of the hopper and the dosing cone disk, the theoretical dosing efficiency of the centrifugal conical disk dispenser will be 376.9 g/s at a minimum rotation frequency of 0.65 rad/s, as opposed to the experimental value, which is 362.36 g/s. At a rotation frequency of the conical disk of 0.65 rad/s, a diameter of the disk from 150 mm, and a height of the annular gap between the discharge throat of the hopper and the dosing conical disk of 10 mm, the theoretical dosing efficiency is 295.0 g/s, whereas in the experiment, it was 278.5 g/s.

A similar difference within 5% is at the angle α of the truncated cone at its base from 10 to 20°.

The use of the theory of design of experiments in empirical studies allows for the establishment of the relationship between the mathematical dependencies and the physical performance of the centrifugal conical disk dispenser. The analysis shows that the system of differential Equation (19) solved by the numerical Runge–Kutta method can be used for different diameters of the conical disk, rotation frequencies, and physical and mechanical characteristics of the bulk material. At the same time, the angle of the truncated cone at its base should be within the range of 10 up to 30°, and the height of the annular gap between the discharge throat of the hopper and the dosing cone disk should be chosen under the condition of preventing spontaneous discharge of the bulk material.

## Figures and Tables

**Figure 1 materials-17-01815-f001:**
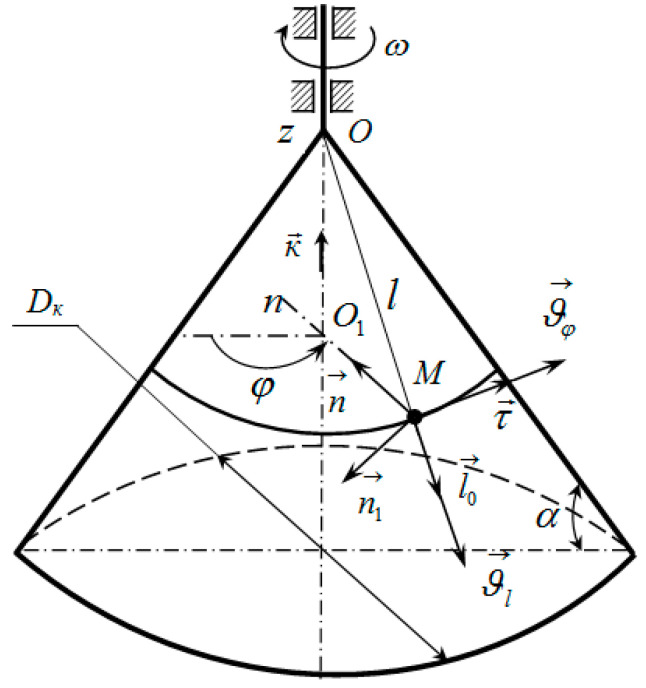
Schematic of velocity vectors of a material point moving along a conical dosing working body under centrifugal rotation: $\vec{n}$—a unit vector directed from point *M* perpendicular to the axis of the cone; ${\vec{n}}_{1}$—a unit vector perpendicular to the surface of the cone.

**Figure 2 materials-17-01815-f002:**
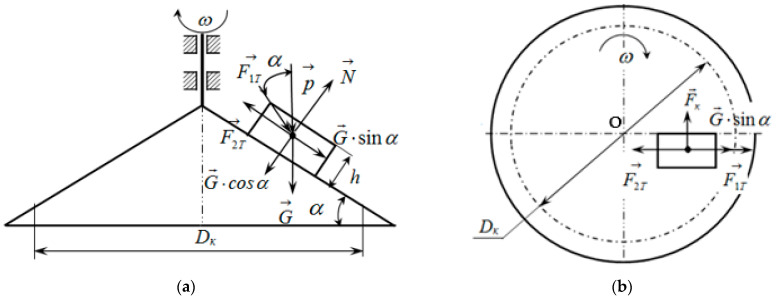
Schematic of the action of forces on the elementary volume of the dispersed material placed on the conical dosing working body: (**a**) cross-sectional view; (**b**) top view.

**Figure 3 materials-17-01815-f003:**
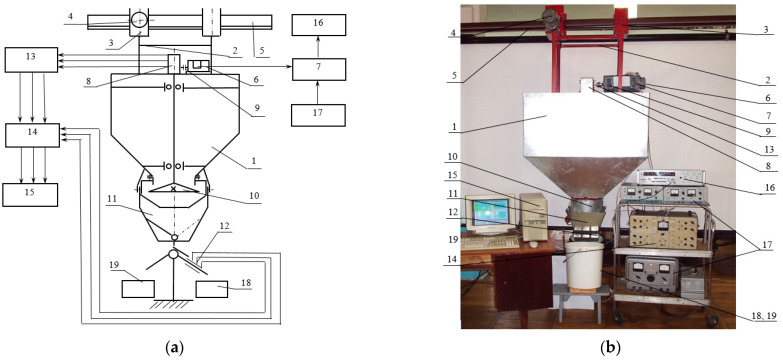
Block diagram (**a**) and general view (**b**) of the laboratory installation of the experimental study of the centrifugal conical disk dispenser of bulk materials: 1—hopper; 2–5—structure of the hopper suspension; 6—conical disk drive; 7—rotational speed meter for the centrifugal conical disk dispenser; 8—reducer; 9—torque sensor; 10—conical disk dispenser; 11—a device for changing the direction of flow of bulk material; 12—bulk material flow meter; 13—torque meter; 14—strain gauge amplifier; 15—computer; 16—electronic frequency meter; 17—DC power supply units; 18, 19—containers for collecting bulk material.

**Figure 4 materials-17-01815-f004:**
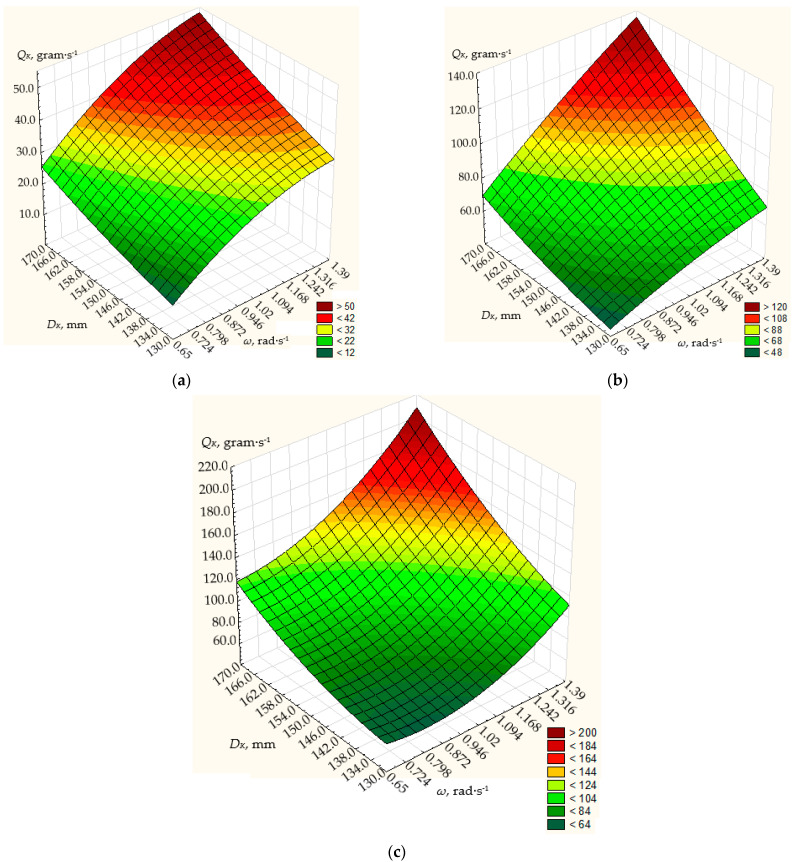
Dependence of the productivity *Q_K_* of the working body of the centrifugal conical disk dispenser on its rotation frequency *n* and diameter *D_K_* at fixed values of the angle of the cone to its base *α* = 10° and the height of the annular gap *h* between the discharge throat of the hopper and the conical disk: (**a**) 6 mm; (**b**) 8 mm; (**c**) 10 mm.

**Figure 5 materials-17-01815-f005:**
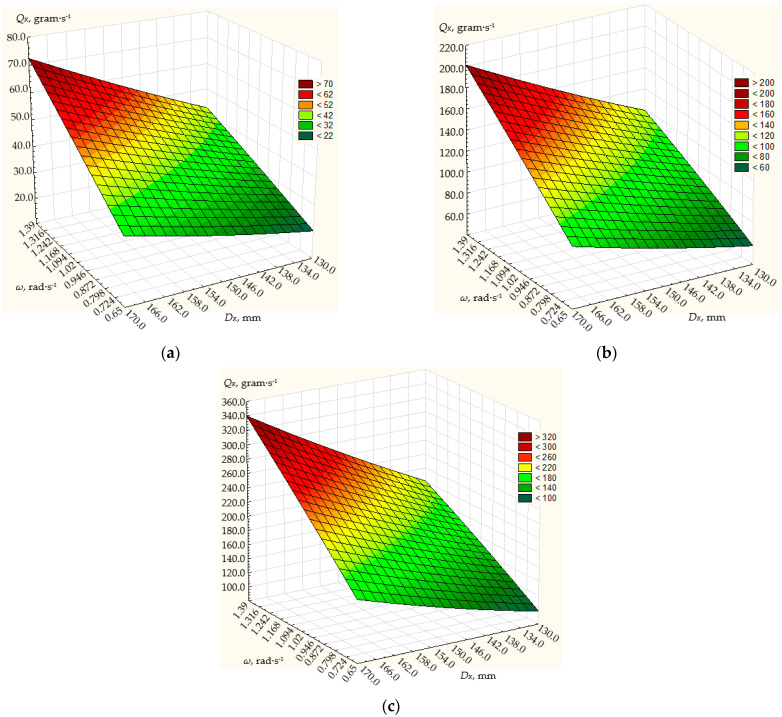
Dependence of the productivity *Q_K_* of the working body of the centrifugal conical disk dispenser on its rotation frequency *n* and diameter *D_K_* at fixed values of the angle of the cone to its base *α* = 20° and the height of the annular gap *h* between the discharge throat of the hopper and the conical disk: (**a**) 6 mm; (**b**) 8 mm; (**c**) 10 mm.

**Figure 6 materials-17-01815-f006:**
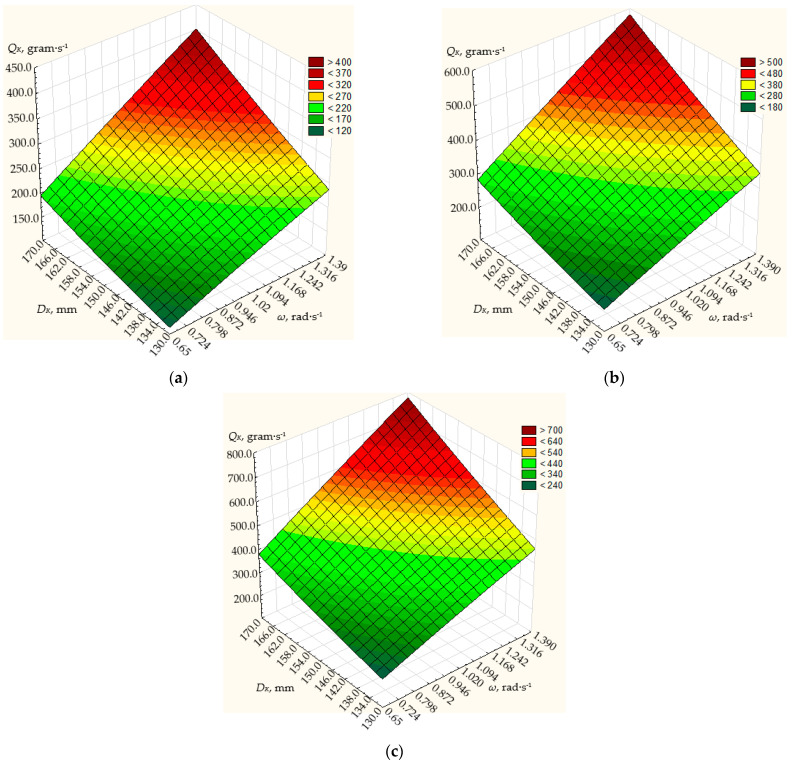
Dependence of the productivity *Q_K_* of the working body of the centrifugal conical disk dispenser on its rotation frequency *n* and diameter *D_K_* at fixed values of the angle of the cone to its base *α* = 30° and the height of the annular gap *h* between the discharge throat of the hopper and the conical disk: (**a**) 6 mm; (**b**) 8 mm; (**c**) 10 mm.

**Figure 7 materials-17-01815-f007:**
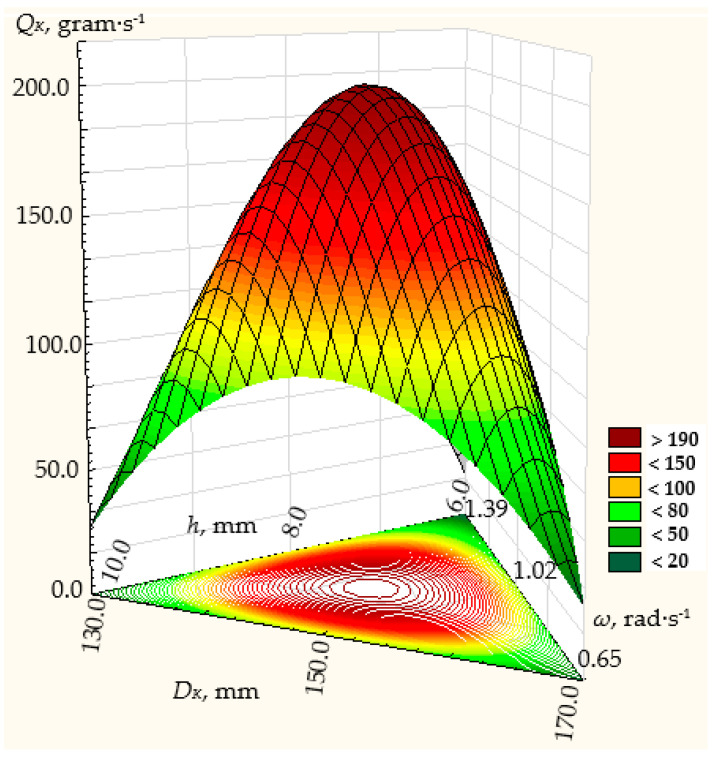
Dependence of the efficiency of the centrifugal conical disk bulk material dispenser at the angle of the cone to its base *α* = 10° on the parameters of the technological process: *ω*—rotation frequency of the disk; *h*—height of the annular gap between the throat of the hopper and the conical disk; *D_K_*—diameter of the disk.

**Figure 8 materials-17-01815-f008:**
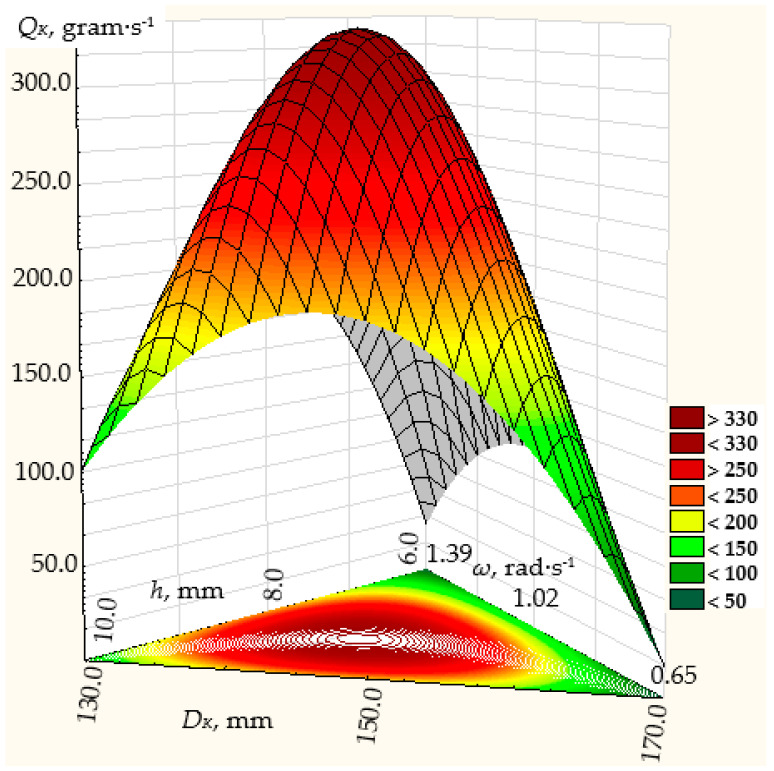
Dependence of the efficiency of the centrifugal conical disk bulk material dispenser at the angle of the cone to its base *α* = 20° on the parameters of the technological process: *ω*—rotation frequency of the disk; *h*—height of the annular gap between the throat of the hopper and the conical disk; *D_K_*—diameter of the disk.

**Figure 9 materials-17-01815-f009:**
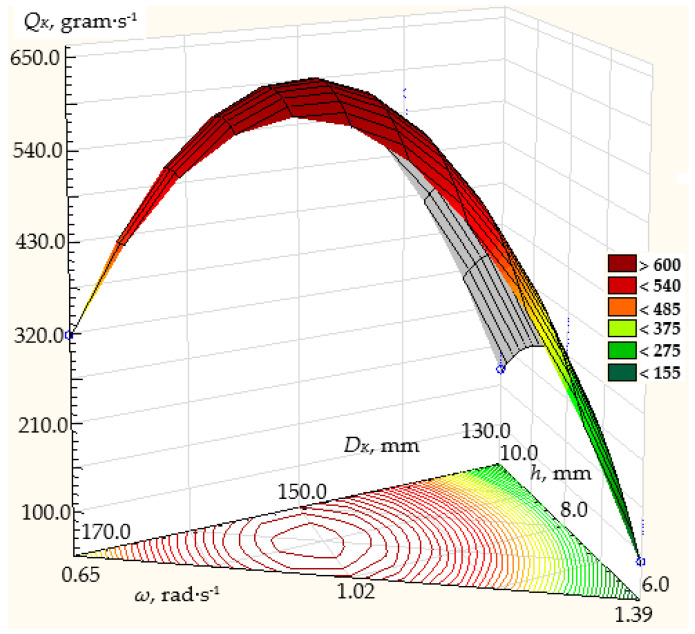
Dependence of the efficiency of the centrifugal conical disk bulk material dispenser at the angle of the cone to its base *α* = 30° on the parameters of the technological process: *ω*—rotation frequency of the disk; *h*—height of the annular gap between the throat of the hopper and the conical disk; *D_K_*—diameter of the disk.

**Table 1 materials-17-01815-t001:** Levels of variation of factors and their coded values in the planned experiment.

Factors	Designation	Dimension	Levels of Factors			Variation Interval
			Upper	Null	Lower	
			Code Values			
			+1	0	−1	
Rotational frequency of the conical disk, *n*	*x* _1_	rad/s	1.39	1.02	0.65	0.37
Height of the annular gap, *h*	*x* _2_	mm	10.0	8.0	6.0	2.0
Diameter of the conical disk, *D_K_*	*x* _3_	mm	170.0	150.0	130.0	20.0

**Table 2 materials-17-01815-t002:** An extended experimental design matrix for a three-factor model of the second order of dependence on the efficiency of the centrifugal conical disk dispenser.

No. of the Experiment	*x* _1_	*x* _2_	*x* _3_	*x*_1_·*x*_2_	*x*_1_·*x*_3_	*x*_2_·*x*_3_	(*x*_1_^′^)^2^	(*x*_2_^′^)^2^	(*x*_3_^′^)^2^	*x*_1_·*x*_2_·*x*_3_	*y*_1_ (*Q_K_*), *α* = 10° [g/s]	*y*_1_ (*Q_K_*), *α* = 20° [g/s]	*y*_1_ (*Q_K_*), *α* = 30° [g/s]
1	2	3	4	5	6	7	8	9	10	11	12	13	14
1	+1	+1	+1	+1	+1	+1	0.3333	0.3333	0.3333	+1	200.00	335.00	745.4
2	0	+1	+1	0	0	+1	−0.6667	0.3333	0.3333	0	138.33	250.46	580.5
3	−1	+1	+1	−1	−1	+1	0.3333	0.3333	0.3333	−1	115.00	165.36	362.36
4	+1	+1	−1	+1	−1	−1	0.3333	0.3333	0.3333	−1	105.67	200.00	449.50
5	0	+1	−1	0	0	−1	−0.6667	0.3333	0.3333	0	68.63	149.73	358.43
6	−1	+1	−1	−1	+1	−1	0.3333	0.3333	0.3333	+1	62.43	97.00	171.46
7	+1	+1	0	+1	0	0	0.3333	0.3333	−0.6667	0	145.00	260.30	601.46
8	0	+1	0	0	0	0	−0.6667	0.3333	−0.6667	0	96.00	200.30	445.40
9	−1	+1	0	−1	0	0	0.3333	0.3333	−0.6667	0	76.53	129.60	278.50
10	+1	−1	+1	−1	+1	−1	0.3333	0.3333	0.3333	−1	50.60	72.36	400.20
11	0	−1	+1	0	0	−1	−0.6667	0.3333	0.3333	0	42.50	52.13	310.00
12	−1	−1	+1	+1	−1	−1	0.3333	0.3333	0.3333	+1	23.90	33.86	180.00
13	+1	−1	−1	−1	−1	+1	0.3333	0.3333	0.3333	+1	31.40	41.53	240.00
14	0	−1	−1	0	0	+1	−0.6667	0.3333	0.3333	0	23.80	30.20	173.30
15	−1	−1	−1	+1	+1	+1	0.3333	0.3333	0.3333	−1	11.40	19.56	112.56
16	+1	−1	0	−1	0	0	0.3333	0.3333	−0.6667	0	40.46	55.40	304.50
17	0	−1	0	0	0	0	−0.6667	0.3333	−0.6667	0	33.80	41.43	226.00
18	−1	−1	0	+1	0	0	0.3333	0.3333	−0.6667	0	14.86	26.00	148.33
19	+1	0	+1	0	+1	0	0.3333	−0.6667	0.3333	0	135.70	199.50	581.4
20	0	0	+1	0	0	0	−0.6667	−0.6667	0.3333	0	95.80	149.50	418.50
21	−1	0	+1	0	−1	0	0.3333	−0.6667	0.3333	0	69.20	97.40	458.60
22	+1	0	−1	0	−1	0	0.3333	−0.6667	0.3333	0	68.80	118.53	353.26
23	0	0	−1	0	0	0	−0.6667	−0.6667	0.3333	0	57.43	88.40	248.50
24	−1	0	−1	0	+1	0	0.3333	−0.6667	0.3333	0	40.60	56.30	163.16
25	+1	0	0	0	0	0	0.3333	−0.6667	−0.6667	0	91.40	156.80	452.46
26	0	0	0	0	0	0	−0.6667	−0.6667	−0.6667	0	81.16	117.20	342.80
27	−1	0	0	0	0	0	0.3333	−0.6667	−0.6667	0	53.40	75.13	218.00
∑	18	18	18	12	12	12	—	—	—	8	—	—	—

**Table 3 materials-17-01815-t003:** The results of the calculations of the coefficients of the regression equation of the dependence of the efficiency of the centrifugal conical disk dispenser on the factors at the angle of the truncated cone to its base *α* = 10°.

Coefficient of the Regression Equation	Coded Coefficient	Real Coefficient
*b* _0_	71.9648	165.5043
*b* _1_	22.3172	106.1478
*b* _2_	40.8261	23.1952
*b* _3_	22.2706	−3.4021
*b* _12_	10.3675	24.9371
*b* _13_	7.23	−1.4815
*b* _23_	13.85	0.0105
*b* _11_	3.4138	−30.4071
*b* _22_	−5.9261	−1.3918
*b* _33_	4.2205	0.0442
*b* _123_	4.3825	0.2961

**Table 4 materials-17-01815-t004:** The results of the calculations of the coefficients of the regression equation of the dependence of the efficiency of the centrifugal conical disk dispenser on the factors at the angle of the truncated cone to its base *α* = 20°.

Coefficient of the Regression Equation	Coded Coefficient	Real Coefficient
*b* _0_	117.9648	164.5176
*b* _1_	41.0672	144.4081
*b* _2_	78.6266	−15.5005
*b* _3_	30.7955	−1.7246
*b* _12_	26.1225	−7.7388
*b* _13_	10.2533	0.5931
*b* _23_	19.7525	0.0045
*b* _11_	−1.0594	−28.1833
*b* _22_	2.3722	−2.0002
*b* _33_	1.8055	0.0621
*b* _123_	6.2637	0.4232

**Table 5 materials-17-01815-t005:** The results of the calculations of the coefficients of the regression equation of the dependence of the efficiency of the centrifugal conical disk dispenser on the factors at the angle of the truncated cone to its base *α* = 30°.

Coefficient of the Regression Equation	Coded Coefficient	Real Coefficient
*b* _0_	380.457	−2815.578
*b* _1_	6.7094	6830.101
*b* _2_	17.9322	577.7693
*b* _3_	−20.5883	1.4054
*b* _12_	−0.5691	−607.203
*b* _13_	−16.8317	−4.9544
*b* _23_	47.3491	0.0867
*b* _11_	−83.1261	−654.011
*b* _22_	−19.8178	−37.1962
*b* _33_	34.7138	−3.2687
*b* _123_	64.605	4.3652

## Data Availability

Dataset available on request from the authors.

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
