# Peer review of "Modeling of the Efficiency of the Centrifugal Conical Disk Dispenser of Bulk Materials"

_materials, 2024, doi:10.3390/ma17081815_

Round 1

Reviewer 1 Report

Comments and Suggestions for Authors

The paper titled "Modelling of the Efficiency of the Centrifugal Disk Cone Dispenser of Bulk Materials" by Vasyl Dmytriv et al. presents an analytical model aimed at understanding and improving the efficiency of centrifugal disk dispensers for bulk materials. The study is well-structured, offering a comprehensive exploration of the mechanical behavior and dynamics of material particles when interacting with a conical centrifugal disk. Through the utilization of the Runge-Kutta numerical method, the paper delivers a systematic approach to solving the differential equations derived from the model, supported by experimental validation to confirm the theoretical findings.

The authors effectively contextualize their research within the broader field by referencing relevant literature, thereby situating their contributions as both a continuation and an advancement of existing knowledge. However, the review identifies a notable gap in the consideration of dynamic stability—a critical aspect in the study of mechanical systems and structures subject to dynamic motion. The paper would significantly benefit from a more thorough theoretical grounding in dynamic stability to enhance its relevance and applicability in practical engineering scenarios.

To address this gap, it is recommended that the authors incorporate references to foundational and recent works specifically focused on the dynamic stability of mechanical systems. Notable examples include "Stability and vibrations of an overcritical speed moving multiple discrete oscillators along an infinite continuous structure" from the European Journal of Mechanics-A/Solids, and "Vibrations and stability analysis of multiple rectangular plates coupled with elastic layers based on different plate theories" from the International Journal of Mechanical Sciences. These references would provide a richer theoretical foundation and underscore the relevance of dynamic stability in the design and analysis of centrifugal disk dispensers.

In conclusion, this paper is a valuable contribution to the field, particularly in its methodological rigor and practical implications for the design of efficient bulk material dispensers. With the suggested enhancements in theoretical depth, particularly regarding dynamic stability, the work could offer even more significant insights and utility for researchers and practitioners alike in mechanical engineering and related disciplines.

Author Response

Reviewer 1

In general:

The paper titled "Modelling of the Efficiency of the Centrifugal Disk Cone Dispenser of Bulk Materials" by Vasyl Dmytriv et al. presents an analytical model aimed at understanding and improving the efficiency of centrifugal disk dispensers for bulk materials. The study is well-structured, offering a comprehensive exploration of the mechanical behavior and dynamics of material particles when interacting with a conical centrifugal disk. Through the utilization of the Runge-Kutta numerical method, the paper delivers a systematic approach to solving the differential equations derived from the model, supported by experimental validation to confirm the theoretical findings.

Dear Reviewer,

Thank you very much for taking the time to carefully read our manuscript. We have accurately read all the comments and referred to all of them. They helped us to significantly improve the article. We have corrected the mistakes, and we hope that now it will meet the standards and receive your recommendations for publication. Below are the general responses to your comments.

Remark  

The authors effectively contextualize their research within the broader field by referencing relevant literature, thereby situating their contributions as both a continuation and an advancement of existing knowledge. However, the review identifies a notable gap in the consideration of dynamic stability—a critical aspect in the study of mechanical systems and structures subject to dynamic motion. The paper would significantly benefit from a more thorough theoretical grounding in dynamic stability to enhance its relevance and applicability in practical engineering scenarios.

To address this gap, it is recommended that the authors incorporate references to foundational and recent works specifically focused on the dynamic stability of mechanical systems. Notable examples include "Stability and vibrations of an overcritical speed moving multiple discrete oscillators along an infinite continuous structure" from the European Journal of Mechanics-A/Solids, and "Vibrations and stability analysis of multiple rectangular plates coupled with elastic layers based on different plate theories" from the International Journal of Mechanical Sciences. These references would provide a richer theoretical foundation and underscore the relevance of dynamic stability in the design and analysis of centrifugal disk dispensers.

Answer

Thank you for your remark.

We analyzed the suggested works and included references to them in our article.

Reviewer 2 Report

Comments and Suggestions for Authors

This paper introduces the modeling of centrifugal dispenser. The paper provides extensive modeling and details, however, my concerns are: 

1. What is the target application? Authors studied the model and experiment. Was the experimental setup based on some practical applications (any reference those setup is from real application).  

For instance, because you mentioned medical application,

>l53 Madadelahi et al. [7] provided a review and analysis of centrifugal microfluidic platforms on disks that dose, separate, and mix various components, both liquid and solid, in microproportions. In medical field, point of care pumping is demanding in microfluidics especially after covid, However, they tend to be slow (doi.org/10.3390/mi7020032, doi.org/10.1063/5.0002169). As such centrifugal pumping is still has challenge (bulky, expensive), but it is traditional and reliable. As such those passive pumping (such as degassed PDMS-based) can be integrated with centrifugal microfluidics to achieve stable performace. So centrifugal dispenser has great potential and advantage, so authors' study is not only useful in the hospital but also bring a wide range of attraction to tackle point-of-care testing for acute disease. On the otherhand, as its dimension is micron size, not only centrifugal force but also other specific dimensionless numbers, physical properties of sample (viscosity etc.) need to be considered in the modeling to match the experimet and modeling. (doi.org/10.1039/C9LC00775J) This is why the centrifugal dispenser in micron scale needs specfic modeling to match with experiment.  

I believe authors did not specify the exact application. The model and experiment seems general (unless the parameters were taken from real application) and maybe good for the journal such as mathematics, but not suitable for Manufacturing Processes and Systems under materials. The author showed some numbers in the abstract, but I have no clue to judge whether this number is valid in the application. I would suggest authors to put some specific applications or just saying where the modeling and experimental setting are taken from assuming what application (which would be smooth to read.)   

2. >L35. The significance of the coefficients in the correlation equation was evaluated using the Student’s test

It is not important who performed the test. 

3. >Table 3. 

Do you have to have so much significant digits in your table? This is issues throughout the manuscript. What is the accuracy of your experiment.

4. >L180 missing a reference.

5. > Fig.2 Could you put where is the rotation center?  

6. In the experimental results, do you have error bars? It was tough to find how many times of experiment authors performed (it says three times)  

7. Do you have some tables compare your work and other reported works (similar to what you wrote before the conclusion section), showing how your model is accurate? 

Overall, authors put lots of efforts on modeling and experiment. However, in my humble opinion, the manuscript does not attract a broad area of reader's attention. I would suggest authors to put some specific applications or just saying where the modeling and experimental setting are taken from assuming what application (which would be smooth to read.) Significant digits and error bars are another big concern. Authors built model in this paper, however, at the same time, the authors did not pay attention to the accuracy of the experiment, numbers and the experiment condition that the proposed model is valid, seems concerns to me. Especially, as authors mention in the introduction, one of the motivations of this paper is to improve dosing accuracy of a centrifugal disk dispenser...

Comments on the Quality of English Language

NA

Author Response

Dear Reviewer,

Thank you very much for taking the time to carefully read our manuscript. We have accurately read all the comments and referred to all of them. They helped us to significantly improve the article. We have corrected the mistakes, and we hope that now it will meet the standards and receive your recommendations for publication. Below are the general responses to your comments.

In general:

This paper introduces the modeling of centrifugal dispenser. The paper provides extensive modeling and details, however, my concerns are: 

Remark 1

What is the target application? Authors studied the model and experiment. Was the experimental setup based on some practical applications (any reference those setup is from real application).  

 For instance, because you mentioned medical application,

>l53 Madadelahi et al. [7] provided a review and analysis of centrifugal microfluidic platforms on disks that dose, separate, and mix various components, both liquid and solid, in microproportions. In medical field, point of care pumping is demanding in microfluidics especially after covid, However, they tend to be slow (doi.org/10.3390/mi7020032, doi.org/10.1063/5.0002169). As such centrifugal pumping is still has challenge (bulky, expensive), but it is traditional and reliable. As such those passive pumping (such as degassed PDMS-based) can be integrated with centrifugal microfluidics to achieve stable performace. So centrifugal dispenser has great potential and advantage, so authors' study is not only useful in the hospital but also bring a wide range of attraction to tackle point-of-care testing for acute disease. On the otherhand, as its dimension is micron size, not only centrifugal force but also other specific dimensionless numbers, physical properties of sample (viscosity etc.) need to be considered in the modeling to match the experimet and modeling. (doi.org/10.1039/C9LC00775J) This is why the centrifugal dispenser in micron scale needs specfic modeling to match with experiment.  

 I believe authors did not specify the exact application. The model and experiment seems general (unless the parameters were taken from real application) and maybe good for the journal such as mathematics, but not suitable for Manufacturing Processes and Systems under materials. The author showed some numbers in the abstract, but I have no clue to judge whether this number is valid in the application. I would suggest authors to put some specific applications or just saying where the modeling and experimental setting are taken from assuming what application (which would be smooth to read.)  

 Answer

Thank you for your remarks. The aim of the study is to substantiate the design parameters and operating modes of the conical working element of the dispenser (the diameter of the dosing working element—the diameter of the conical disk, the annular gap between the discharge throat of the hopper and the conical disk, the rotational speed of the conical disk), which affect dosing productivity. The angle of the conical dosing disk was a constant parameter. The bulk material is real. It is a mixture of crushed grains, flour (as a mixture of crushed grains, including corn, buckwheat, and others). The material mixture was selected so that its density was 550 kg/m3. The moisture content of the bulk material (mixture) was 13%, the angle of natural repose was 33°, the coefficient of sliding friction of the bulk material f = 0.2-0.25, the coefficient of internal sliding friction (adhesion) f1 = 0.65, the external coefficient of sliding friction between the bulk material and the conical dosing disk f2 = 0.443.

Remark 2

>L35. The significance of the coefficients in the correlation equation was evaluated using the Student’s test

It is not important who performed the test. 

Answer

Student’s t-test is a common statistical tool, and the calculated values of the Student's coefficient were determined for the general regression model using the Statistics 12.0 software. The tabular value of the Student's coefficient is 1.7138 for a significance level of 0.05. Accordingly, the calculated values of the Student's coefficient for the regression models, formulas (26), (27), (28), are 24.522, 24.638, 11.84, which are higher than the tabular values.

Remark 3

 >Table 3. 

Do you have to have so much significant digits in your table? This is issues throughout the manuscript. What is the accuracy of your experiment.

Answer

In Table 3, the coefficients of the equation are presented with four decimal places after the whole number to enhance the precision of determining the Student's criterion.

Remark 4

>L180 missing a reference.

Answer

Corrected. Square brackets removed—it was a typo.

Remark 5

> Fig.2 Could you put where is the rotation center?  

Answer

Corrected. The center of rotation in Fig. 2 is marked with the letter O.

Remark 6

In the experimental results, do you have error bars? It was tough to find how many times of experiment authors performed (it says three times)  

Answer

Twenty-seven experiments were conducted with each experiment repeated three times. Table 2 shows the average value of the response criterion—dosing efficiency. The article does not give the value of each individual experiment, but rather the average value. We believed that there was no need to provide the matrix of the experimental results, as this would significantly increase the length of the article.

Remark 7

Do you have some tables compare your work and other reported works (similar to what you wrote before the conclusion section), showing how your model is accurate?

Answer

In papers [4-7], researchers describe the kinematics of the movement of material particles on a disk. Similarly, in [14-18, 27-28], mathematical models of particle motion were considered, as well as the uniformity of their distribution across the surface when fertilizers were applied.

The authors did not study the performance of disk pipettes in terms of dosing accuracy. Therefore, it is not possible to compare the accuracy performance parameter with other studies cited in the article.

Reviewer 3 Report

Comments and Suggestions for Authors

The research presented in the paper "Modelling of the Efficiency of the Centrifugal Disk Cone Dispenser of Bulk Materials" addresses an important aspect of industrial processes, particularly in fields where precise dosing of bulk materials is crucial. Here's a review of the various components of the study:

1. The abstract could be improved by emphasizing the novelty and contribution of the research. This would help readers quickly understand the unique aspects of the study.

2. The assumptions made in the derivation of the analytical model, such as the constant bulk pressure, could be re-examined for potential refinement. Additionally, it would be beneficial to provide clear justifications for the numerical values of the variables f, f1, and f2 used in the analysis.

3. The paper mentions that the experiment was repeated five times at the same levels of factors. This is a good practice as it helps to ensure the reliability and reproducibility of the results. However, it would be beneficial to provide more details about the experimental setup and procedure.

4. Please provide a rationale for the selection of the three parameters and levels in Table 2. Also, considering that Qk is a strictly increasing or decreasing function, a discussion on why the values of the parameters can’t be adjusted further up or down would be insightful.

5. Please specify the type of material or particles that correspond to a bulk density of 550 kg/m³. Also, discuss the applicability of the study’s findings to different densities or particles.

6. Please clarify the conditions under which the theoretical results differ from experimental data by no more than 5%.

7. The Conclusion states that the results of experimental studies agree with the results of theoretical simulations. This is a strong point, as it validates the theoretical model with empirical evidence. However, the conclusions could be expanded to discuss the implications of these findings for the field and potential future research directions.

8. Ensure that abbreviations like "DEM" are spelled out upon first usage.

9. Verify the completeness of the sentence in Line 180, where it mentions "dependencies of the kinematics [], we..."

10. Correct the unit in Line 314 from "s-1" to "rad s-1" as suggested.

11. Confirm the accuracy and consistency of terms like "an angular velocity ω" and "Rotational frequency of the conical disk, n" in Line 184.

12 Coordinate the placement of Tables 3-5 to appear on Page 11 upon their first mention.

Author Response

In general:

The research presented in the paper "Modelling of the Efficiency of the Centrifugal Disk Cone Dispenser of Bulk Materials" addresses an important aspect of industrial processes, particularly in fields where precise dosing of bulk materials is crucial. Here's a review of the various components of the study:

Dear Reviewer,

Thank you very much for taking the time to carefully read our manuscript. We have accurately read all the comments and referred to all of them. They helped us to significantly improve the article. We have corrected the mistakes, and we hope that now it will meet the standards and receive your recommendations for publication. Below are the general responses to your comments.

Remark 1

The abstract could be improved by emphasizing the novelty and contribution of the research. This would help readers quickly understand the unique aspects of the study.

Answer

Thank you for your valuable feedback on our research article. We have addressed the issue, and the necessary modifications have been made to the article. We have added the following at the end of the abstract: “The obtained system of differential equations makes it possible to model the radial velocity of the ascent of bulk material from the conical rotating disk depending on the rotation frequency, disk diameter, and the height of the annular gap between the discharge throat of the hopper and the conical disk. The analytical model enables modeling the productivity of the conical dispenser for bulk materials for arbitrary parameters of rotation frequency, disk diameter, and the size of the annular gap between the discharge throat of the hopper and the conical disk.”

Remark 2

The assumptions made in the derivation of the analytical model, such as the constant bulk pressure, could be re-examined for potential refinement. Additionally, it would be beneficial to provide clear justifications for the numerical values of the variables ff1, and f2 used in the analysis.

Answer

The volumetric pressure acts on the conical disk dispenser, which is determined by formula (9)—this is the pressure of an elementary volume, as indicated in Fig. 2a. The variability of the volumetric pressure is compensated by the force of internal sliding friction between elementary volumes through the coefficient of internal sliding friction, formula (12). The bulk material is real. It is a mixture of crushed grains, flour (as a mixture of crushed grains, including corn, buckwheat, and others). The material mixture was selected so that its density was 550 kg/m3. The moisture content of the bulk material (mixture) was 13%, the angle of natural repose was 33°, the coefficient of sliding friction of the bulk material f = 0.2-0.25, the coefficient of internal sliding friction (adhesion) f1 = 0.65, the external coefficient of sliding friction between the bulk material and the conical dosing disk f2 = 0.443.

Remark 3

The paper mentions that the experiment was repeated five times at the same levels of factors. This is a good practice as it helps to ensure the reliability and reproducibility of the results. However, it would be beneficial to provide more details about the experimental setup and procedure.

Answer

Regarding the fivefold repetition of the experiment, perhaps this is an inaccuracy. In the article, lines 260–262: “To study the influence of these factors, according to the matrix of the experiment (Table 2), the experiment was carried out three times according to the previously mentioned method [43]”. In line 407, it was corrected to three.

The laboratory setup shown in Fig. 3 is an experimental design. The parameters were measured using rotation frequency sensors for the disk conical dispenser, while the productivity was measured by a bulk material flow meter, with a load cell as the recording element. The information was read by a computer through the L-154 analog-to-digital module. The experimental procedure was classical.

Remark 4

Please provide a rationale for the selection of the three parameters and levels in Table 2. Also, considering that Qk is a strictly increasing or decreasing function, a discussion on why the values of the parameters can’t be adjusted further up or down would be insightful.

Answer

The following parameters were selected: the rotation frequency of the conical disk, the disk diameter, and the size of the annular gap between the discharge throat of the hopper and the conical disk. These parameters are adjustable. The design parameters of the dispenser are the diameter of the conical disk and the size of the annular gap between the discharge throat of the hopper and the conical disk. The technological parameter is the rotation frequency of the conical disk. The levels of these parameters. The annular gap between the discharge throat of the hopper and the conical disk is chosen based on the condition of preventing self-discharge of the bulk material (maximum gap) and allowing the material to exit during disk rotation (minimum gap). Each specific bulk material has its own limits. The cone diameter was chosen according to the design of the dispenser and the technical capabilities of its production. However, this is a real machine for dosing bulk material. The rotation frequency was selected taking into account the technical feasibility of measuring the productivity parameter. The research objective here is to investigate experimentally the reliability of the analytical model of the productivity of the disk cone dispenser.

Remark 5

Please specify the type of material or particles that correspond to a bulk density of 550 kg/m³. Also, discuss the applicability of the study’s findings to different densities or particles.

Answer

The bulk material is real. It is a mixture of crushed grains, flour (as a mixture of crushed grains, including corn, buckwheat, and others). The material mixture was selected so that its density was 550 kg/m3. Confirmation of the reliability of the analytical model of the productivity of the disk conical dispenser at a density of 550 kg/m³ allows us to assert that the analytical model of the productivity of the conical disk dispenser for bulk materials is operational and applicable to other bulk materials with different densities.

Remark 6

Please clarify the conditions under which the theoretical results differ from experimental data by no more than 5%.

Answer

The error of 5% between theoretical and experimental data indicates that the analytical model in the form of a system of differential equations takes into account the parameters and forces that act in the real physical process of dosing with such a dispenser. In fact, the mathematical model is original and has never been seen before. It is a meticulously developed system of differential equations taking into account all the components of forces acting on the bulk material during dosing with the rotating conical disk.

Remark 7

The Conclusion states that the results of experimental studies agree with the results of theoretical simulations. This is a strong point, as it validates the theoretical model with empirical evidence. However, the conclusions could be expanded to discuss the implications of these findings for the field and potential future research directions.

Answer

Thank you for your remark. We are working on topics for future research in this area.

Remark 8

Ensure that abbreviations like "DEM" are spelled out upon first usage.

Answer

Thank you for pointing that out. It has now been spelled out—discrete element method (DEM).

Remark 9

Verify the completeness of the sentence in Line 180, where it mentions "dependencies of the kinematics [], we...".

Answer

Corrected. The square brackets have been removed; it was a typo.

Remark 10

Correct the unit in Line 314 from "s-1" to "rad s-1" as suggested.

Answer

Corrected.

Remark 11

Confirm the accuracy and consistency of terms like "an angular velocity ω" and "Rotational frequency of the conical disk, n" in Line 184.

Answer

The rotation of a three-dimensional rigid body is described by the angular velocity vector, ω = dφ/dt, as well as the angular acceleration: d2φ/dt2. The peculiarity of the mathematical model of productivity lies in the application of the acceleration vector of a material point taking into account linear and angular accelerations, formula (7).

Remark 12

Coordinate the placement of Tables 3-5 to appear on Page 11 upon their first mention.

Answer

Thank you for your remark. We believe that this is not feasible. The results of the experiment are described in section 3.2.

Round 2

Reviewer 2 Report

Comments and Suggestions for Authors

Thank you for your detailed explanation. The manuscript was improved. Please consider below.

Thank you for your remarks. The aim of the study is to substantiate the design parameters and operating modes of the conical working element of the dispenser (the diameter of the dosing working element—the diameter of the conical disk, the annular gap between the discharge throat of the hopper and the conical disk, the rotational speed of the conical disk), which affect dosing productivity. The angle of the conical dosing disk was a constant parameter. The bulk material is real. It is a mixture of crushed grains, flour (as a mixture of crushed grains, including corn, buckwheat, and others). The material mixture was selected so that its density was 550 kg/m3. The moisture content of the bulk material (mixture) was 13%, the angle of natural repose was 33°, the coefficient of sliding friction of the bulk material f = 0.2-0.25, the coefficient of internal sliding friction (adhesion) f1 = 0.65, the external coefficient of sliding friction between the bulk material and the conical dosing disk f2 = 0.443.

-> Please include this in the method section.

The article does not give the value of each individual experiment, but rather the average value. We believed that there was no need to provide the matrix of the experimental results, as this would significantly increase the length of the article.

-> You may include in the supplemental information.

The authors did not study the performance of disk pipettes in terms of dosing accuracy. Therefore, it is not possible to compare the accuracy performance parameter with other studies cited in the article.

-> I believe the accuracy of the experiment should be the significant digit.

>In conclusion section, for example, L500 "the theoretical dosing productivity is 180.1 g/s, and in the experiment—171.5 g/s" 

-> Where does the 171.5g/s come from. Please consider discuss in the main body.  

Comments on the Quality of English Language

na

Author Response

Reviewer 2

Dear Reviewer,

Thank you very much one again for all the comments and suggestions. We did our best to implement all the suggested changes.

Remark 1

Thank you for your detailed explanation. The manuscript was improved. Please consider below.

Thank you for your remarks. The aim of the study is to substantiate the design parameters and operating modes of the conical working element of the dispenser (the diameter of the dosing working element—the diameter of the conical disk, the annular gap between the discharge throat of the hopper and the conical disk, the rotational speed of the conical disk), which affect dosing productivity. The angle of the conical dosing disk was a constant parameter. The bulk material is real. It is a mixture of crushed grains, flour (as a mixture of crushed grains, including corn, buckwheat, and others). The material mixture was selected so that its density was 550 kg/m3. The moisture content of the bulk material (mixture) was 13%, the angle of natural repose was 33°, the coefficient of sliding friction of the bulk material f = 0.2-0.25, the coefficient of internal sliding friction (adhesion) f1 = 0.65, the external coefficient of sliding friction between the bulk material and the conical dosing disk f2 = 0.443.

-> Please include this in the method section

Answer

Thank you for your remark. It has now been included in the Materials and Methods section.

Remark 2

The article does not give the value of each individual experiment, but rather the average value. We believed that there was no need to provide the matrix of the experimental results, as this would significantly increase the length of the article.

-> You may include in the supplemental information.

Answer

The number of experiments is 243. The average value of the experiment results is also informative. The results were evaluated according to the Student and Fisher criteria. Still, should anyone need to see the individual results, we have added the following data availability statement: “Owing to the extensive nature of the empirical data, the individual results are not included in the text of the article; however, the complete results of all the experiments performed in this study can be made available on request from the corresponding authors.”

Remark 3

The authors did not study the performance of disk pipettes in terms of dosing accuracy. Therefore, it is not possible to compare the accuracy performance parameter with other studies cited in the article.

-> I believe the accuracy of the experiment should be the significant digit.

 >In conclusion section, for example, L500 "the theoretical dosing productivity is 180.1 g/s, and in the experiment—171.5 g/s" 

-> Where does the 171.5g/s come from. Please consider discuss in the main body.  

Answer

This serves as an example for specific dosing parameters. With an angle of the truncated cone at its base of 30°, a rotation frequency of the conical disk of 0.65 rad/s, a diameter of the disk from 130 mm, and a height of the annular gap between the discharge throat of the hopper and the dosing conical disk 10 mm, the theoretical dosing efficiency is 180.1 g/s, while during the experiment it reaches 171.5 g/s.

This illustrates the specific values obtained during theoretical efficiency calculation and the efficiency achieved during the experiments. The difference between the theoretical and experimental values is 8.6 g/s—the absolute deviation. Expressed as percentage, the error is 8.6/180.1 = 4.78%.

Similarly, for the other operating parameters of the dispenser, deviations do not exceed 5%.

We investigated how the theoretical determination of dispenser efficiency differs from the obtained value during experimentation, and to what extent theory corresponds to the actual efficiency of the dispenser. Accordingly, the accuracy of the theoretical calculation is at least 95%. The essence of the study lies in the correspondence between theory and experiment.